Review article

# Towards precision medicine using biochemically triggered cleavable conjugation
Badri Parshad [1,2,9], Smriti Arora[3,9], Balram Singh[4], Yuanwei Pan[5], Jianbin Tang[6], Zhigang Hu [7] &
Hirak K. Patra [8] ✉

Personalised and precision medicines are emerging as the future of therapeutic strategies. Biochemically triggered cleavable conjugation is thus crucial and timely due to its potential to response as per the loco-regional environment. It enables targeted release of therapeutic agents in response to specific biochemical signals and thus minimizing off-target effects and improving treatment precision. It holds promise in a range of biomedical applications, including cancer therapy, senolytic therapy, gene therapy, and regenerative medicine. The focus of this review is to offer comprehensive insight into the significance of biochemically cleavable conjugations within intrinsically stimuli-responsive architectures. Pathological conditions and alteration in tissues microenvironment in the body exhibit distinct biochemical settings characterized by change in redox potential, pH level, hypoxia, reactive oxygen species (ROS), and various catalytic protein/enzyme overexpression. Understanding these intrinsic features is crucial for researchers aiming to develop intelligent cleavable bio-engineered systems for biomedicines. By strategically designing cleavable linkage, researchers can leverage the variations in the tumor, infection, inflammation, and senescence microenvironments. Through an extensive examination of relevant literature, we present a comprehensive classification of the intrinsic physicochemical differences found in pathological areas and their applications in drug delivery, prodrug activation, imaging, and theranostics for future personalised medicines. This review will provide comprehensive guidance and critical insights to researchers in both industry and academia who are involved in the design of advanced, functional biochemically cleavable conjugations.

A better exploration of locoregional intrinsic changes in the microenvironment of pathological areas, such as tumors, infections, inflammations, and cellular senescence, allowed to develop strategies those are specifically reactive to certain deviant conditions of affected tissues[1–3]. These strategies are useful for a variety of biomedical applications, including drug delivery, imaging, sensing, and theranostics as they can "switch-on" certain functions upon exposure to such intrinsic incitements, such as cargo release from a carrier, enhanced fluorescence output, and higher affinity towards

specific biomolecules[4]. The biochemistry of various affected areas in collaboration with precise stimuli responsive chemical entities helps to engineer such temporal and spatial biomaterial designing. Altered biochemical environment such as redox potential, pH, hypoxia, temperature, reactive oxygen species (ROS) and overexpression of proteins/enzymes represent the intrinsic characteristics of affected pathological areas/tissues in the body[5–11]. Notably, in comparison to external stimuli, the intrinsic stimuli offer unique advantages as they do not need an external device and can precisely be

[1]Wellman Center for Photomedicine, Massachusetts General Hospital, Harvard Medical School, Boston, MA, USA. [2]Institute of Nano Medical Sciences, University of Delhi, Delhi, India. [3]Institut für Chemie und Biochemie Organische Chemie, Freie Universität Berlin, Takustr. 3, Berlin, Germany. [4]Faculty of Science and Engineering, Swansea University, Swansea, UK. [5]Institute of Biomedical Health Technology and Engineering, Shenzhen Bay Laboratory, Shenzhen, China. [6]Zhejiang Key Laboratory of Smart Biomaterials and College of Chemical and Biological Engineering, Zhejiang University, Hangzhou, China. [7]Center for Hydrogen Science, School of Material Science and Engineering, Shanghai Jiao Tong University, Shanghai, China. [8]Department of Surgical Biotechnology, UCL Division of Surgery and Interventional Science, University College London, London, UK. [9]These authors contributed equally: Badri Parshad, Smriti Arora.
✉e-mail: hirak.patra@ucl.ac.uk

controlled from tissue to the cellular organelle level. Herein, we have comprehensively outlined physicochemical distinctions in the tumor, infection, inflammation, and senescence microenvironment, utilized by the researchers to develop smart cleavable bio-engineered systems of interest through rationally designed cleavable bonds. In this review, particular emphasis has been given to the biologically cleavable chemical conjugations as smart strategies for constructions of advanced architectures for biomedical biotechnological applications such as drug delivery, prodrug activation, imaging and theranostics. External stimuli responsive and non-cleavable internal stimuli responsive conjugations are beyond the scope of this review. For the comprehensive knowledge of external-stimuli viz light, thermal, ultrasound, magnetic field, and electric field responsive systems, and non-cleavable systems viz charge inversion, polarity variation, expansion and shrinking, see the review articles by Deng et al.[12], Ray et al.[13], Wei et al.[14], Fischer et al.[15], Li et al.[16], Mirvakili et al.[17], Yu et al.[18], and Kolosnjaj-Tabi et al.[19].

## Intrinsic physiochemical distinctions in the tumor microenvironment

Tumor microenvironment offers diverse characteristics, including but not limited to (i) tumor acidity (pH 6.5–7.2); (ii) acidic pH in the endosome (pH 5.5–6.8) and lysosome (pH 4.5–5.5); (iii) tumor extracellular enzymes such as proteases, phosphatases, and glycosidases; (iv) elevated glutathione levels in the cytosol and cell nucleus; (v) ROS in the mitochondria; (vi) degenerative enzymes in the lysosomes[20]. All these dynamic alterations at the cellular and tissue level are a result of nonstandard growth and variations in cells, the extracellular matrix (ECM), and the blood vessels. Moreover, these mentioned characteristic alterations are inter-related, with bi-directional feedback, for example, high proliferation rate of growing solid tumor mass would exhaust the local supply of $O_2$ sooner. A low-oxygen (hypoxic) environment triggers changes in cellular behavior as cells must adapt to survive. These adaptations are regulated by hypoxia-inducible factor-1 (HIF-1), which controls various biological pathways. When oxygen levels are low, HIF-1α stabilizes, activating downstream signaling that influences processes such as apoptosis, cell cycle arrest, angiogenesis, glycolysis, and pH adaptation[21]. Additionally, the rapid proliferation of cancer cells leads to increased tumor mass within a confined space, raising intra-tumoral pressure and compressing blood and lymphatic vessels[22]. This restricts oxygen delivery and waste removal, contributing to a hypoxic and acidic tumor microenvironment[23–25]. The acidity primarily results from anaerobic glycolysis and lactic acid production[26]. While metabolic changes due to hypoxia (the Pasteur effect) play a role, tumors often favor anaerobic glycolysis even in the presence of oxygen, a phenomenon known as the Warburg effect. A similar inter-relation at the molecular level is considered between glutathione (GSH) concentration and the cellular reducing environment[27]. GSH controls the cellular reducing environment mainly through the formation of disulfide bonds and the reaction with excess ROS. GSH is a crucial antioxidant within the cell, synthesized from glutamine-derived carbons. Under oxidative stress, characterized by elevated reactive oxygen species (ROS), GSH in its reduced form is increasingly converted into its oxidized form, GSSG. This oxidized state can be harmful to the cell, as GSSG functions as a pro-oxidant. Also, some of the classical tumor promoters also activate GSH synthesis[28]. It has been reported that the GSH concentration in many cancers including breast, ovarian, lung, as well as head and neck cancers is at least 4-fold higher than that in normal tissues[21,29,30]. The exact changes in GSH concentration in different[31] cancer types are detailed in Table 1.

## Chemical environment in cancer and its relation to infection, chronic inflammation, and cellular senescence

In the last decades, Rudolf Virchow's hypothesis[32] has been supported by number of evidence that various cancers are triggered by infection and chronic inflammation (Fig. 1)[33]. Generally, inflammation is a beneficial response activated to restore tissue injury and pathogenic agents. However, unregulated inflammation becomes chronic, inducing malignant cell transformation in the surrounding tissue. The inflammatory response generates reactive oxygen species (ROS) and reactive nitrogen species (RNS), along with bioactive signaling molecules such as cytokines, growth factors, and chemokines. These components support sustained cell proliferation, enhance survival signals to prevent apoptosis, and promote angiogenesis. Additionally, extracellular matrix-modifying enzymes like metalloproteinases contribute to epithelial-mesenchymal transition (EMT), facilitating cancer development[34]. Also, the inflamed tissues share various intrinsic biochemical stimuli with the carcinogenic tissues, in comparative to the healthy tissues, such as oxidative stress, acidic pH, and overexpressed enzymes enabling the development of stimuli responsive nanoengineered systems.

Moreover, cancer closely interrelates with aging as several molecular pathways and underlying causes of aging significantly overlap with those of cancer. One of the key phenomena of aging, cellular senescence, relates aging with cancer. Senescence is generally regarded as a tumor suppressive process, preventing cancer cell proliferation and suppressing malignant progression. However, there are evidence that suggest senescence have a tumor promoting effect, owing to the proangiogenic influence of some components of senescence-associated secretory phenotype (SASP) such as vascular endothelial growth factor (VEGF) or the impact of senescent fibroblasts on adjacent tumor cells[35,36]. Moreover, there are also evidences for "immune senescence", which associates the aging of immune system with failure of immune surveillance and contribution to cancer progress in older people[37]. One primary characteristic of senescent cells is elevated activity of lysosomal β-galactosidase, exploited as senescence associated β-galactosidase (SA-β-gal)[38]. Remarkably, SA-β-gal is one of the widely used markers for identifying senescence in vitro and in vivo, which is linked to increased level of lysosomes[39–41].

## Chemical strategies for intrinsically cleavable systems

The highly specific delivery or activation of the desired targets begins with introducing critical cleavable linkers into a variety of platforms such as small molecules, polymers, antibody-drug conjugates (ADCs), and nanoparticles, during synthetic stage. These linkages then interact with specific intrinsic changes at the pathological regions to ensure effective delivery or activation of the system for the application of interest. Such cleavable chemistries have been detailed in Fig. 1. However, such a selectively reactive linker technology must strive towards these key properties: (a) high stability in circulation; (b) preserve the activity of the specific components such as antibody, cargo, proteins, fluorescent tags, and biomolecules; (c) high sensitivity towards physiochemical changes; (d) non-hemolytic; (e) should not generate any immunological response; (f) high cell permeability; (g) should afford controlled changes in the macro-scale structure of the whole system; (h) should not accumulate in body organs/tissue above cytotoxic concentration; (i) length of the linker used for conjugation when dealing with the targeting ligands as the linker enhances the proximity, and probability of the ligand-target interactions.

### Enzyme-responsive cleavable conjugations

Enzymes play a fundamental role in most physiological processes, such as cell proliferation, apoptosis, angiogenesis, migration, and autophagy. Certain enzymes are highly expressed in diseased organs, while they are not expressed or expressed at relatively low levels in normal tissues[8,38,42]. The change in enzyme expression has become the most promising trigger to develop stimuli-responsive systems, owing to their exceptional selectivity, lack of side-effects and mild reactions involved[43]. Systems constituting special chemical linkages like peptides, esters, glycosidic bonds, disulfides, or azo bonds can undergo physical or chemical degradation in response to changes in the expression of specific enzymes[44,45]. These systems can be used in a variety of biomedical applications, such as cargo delivery, imaging, prodrug activation, and theranostics[44,46,47].

**Peptide linkage**. Peptide linkages containing enzyme-responsive systems can be created using diverse materials like polymers, amphiphiles,

**Table 1 | Different physiochemical distinctions at various tumor sites**

| Physiochemical signals | Normal tissues | Tumor tissues | Tumor models | Ref |
|---|---|---|---|---|
| pH | 7.4 (human plasma) | ~6.8 (tumor extracellular environment), 4.3–5.2 in the endosomes and lysosomes | H9618a cells, U937 cells | 155 |
| GSH | 1.2 mM, 1300–2700 nmol/g-tissue wet weight | 0.5–3 mM, 600–4600 nmol/g-tissue wet weight | Brain Tumors | 31 |
| | 70 – 250 nmol/g-tissue | 250–2000 nmol/g-tissue wet weight | Breast | |
| | 44 nmol/mg-protein | 5700 nmol/g-tissue wet weight, 10–50 nmol/mg-protein | Colorectal cancers | |
| | 5.52 ± 3.27 µmol/mg-protein | 6–4600 nmol/mg-protein | Liver Cancer | |
| | 5–50 nmol/mg-protein | 0.3–126 nmol/mg-protein | Ovarian | |
| ROS | ~3 µM (human plasma) | 50–100 µM (extracellular) | HeLa cells | 156 |
| Hypoxia | 9.5%–4.6%* *% refers to oxygen concentration | 1.7% (Brain), 1.5% (Breast), 1.2% (Cervical), 1.3% (Kidney Cortex), 0.8% (Liver), 2.2% (Lung), 0.3% (Pancreas), 1.8% (Rectal mucosa) | Brain, Breast, Cervical, Renal, Liver, NSCLC, Pancreatic, Rectal Carcinoma | 6 |

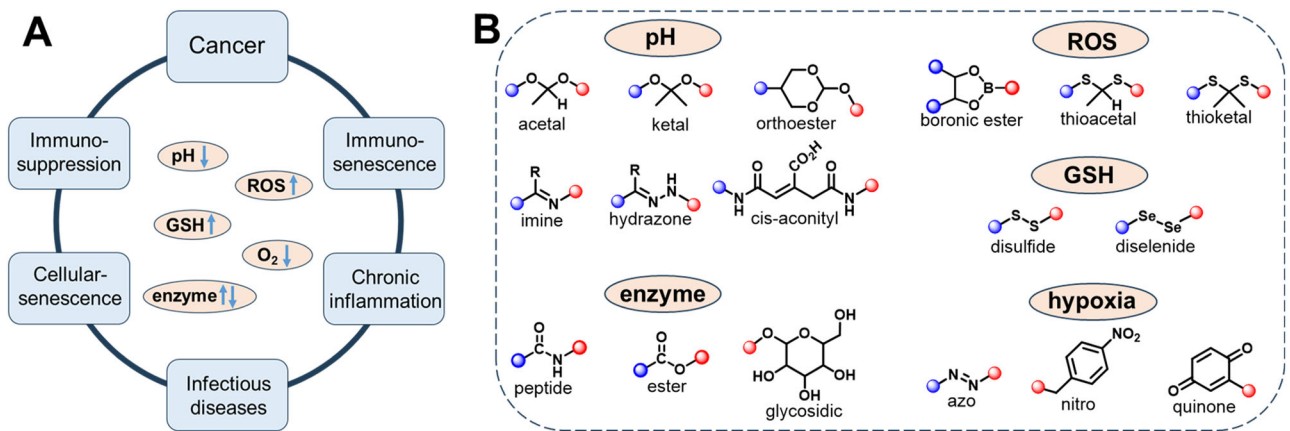

**Fig. 1 | Variation in the concentration of biochemical species and their corresponding responsive linkages. A** The relationship between cancer and other disorders. **B** Biochemical-specific strategies for cleavable linker.

nanoparticles, and hydrogels. These materials can be incorporated with peptides cleaved by specific enzymes, such as trypsin and cathepsin, leading to selective drug or imaging agent activation/release. An interesting approach involves using a cathepsin-cleavable valine-citrulline (Val-Cit) linker in a macromolecular prodrug platform for pulmonary intracellular infections, providing controlled drug release and enhanced antibacterial efficacy (Fig. 2A). Tandem cleavable linkers prodrug platform utilizing Val-Cit linkage chemistry has been reported by Su et al. against pulmonary intracellular infections (Fig. 2B)[48]. This prodrug (ciprofloxacin) incorporates two advantageous features, a protease (cathepsin B) cleavable Val-Cit linker, and multivalent mannose ligands to target and enhance internalization by alveolar macrophage. The design has shown improved stability, tolerability, and efficacy in antibody-drug conjugates, addressing issues with premature cleavage by extracellular enzymes[49,50].

Recent studies have highlighted challenges of often prematurely cleaved antibody-drug conjugates by extracellular enzymes e.g., elastase, leading to systemic release of cytotoxic. Chuprakov et al. tried to address this limitation through the development of a tandem cleavable linker which allows two sequential enzymatic cleavage events to mediate controlled payload release[49]. They modified the Val-Cit-p-aminobenzylcarbamate (PABC) linker technology developed by Seagen which has been employed on three of the nine approved antibody-drug

conjugates and tested on more than 20 clinically relevant molecules[50]. The Val-Cit-PABC (Cathepsin B cleavable) was utilized in combination with glucuronide moiety (β-Glucuronidase cleavable). The dipeptides are protected from degradation in circulation by sterically encumbering glucuronide entity which upon internalization and lysosomal degradation, removes the monosaccharide, resulting in degradation of the exposed dipeptide, and thereby, releasing off the attached payload monomethyl auristatin E inside the target cell.

Moreover, to diversify the portfolio of enzyme-cleavable linkers, Miller et al. screened a panel of 75 peptide FRET pairs for cleavage in plasma and lysosomal extracts[51]. Surprisingly, a series of asparagine-containing peptides were found to be cleaved more quickly than Val-Cit-type linkers while retaining excellent stability in plasma. The peptides could be cleaved by legumain, an asparaginyl endopeptidase which is overexpressed in a variety of cancers and is present in the lysosome. Importantly, the Asn-containing linkers were completely stable to human neutrophil elastase. Several other peptide linkers such as glutamic acid-glycine-citrulline and glutamic acid-valine-citrulline have been developed to be utilized in antibody-drug conjugate as the highly selective and quickly hydrolysable alternate of Val-Cit[52,53].

**Ester linkage.** Crafting polymeric and amphiphilic systems with functional groups readily responsive to lipases has attracted increasing

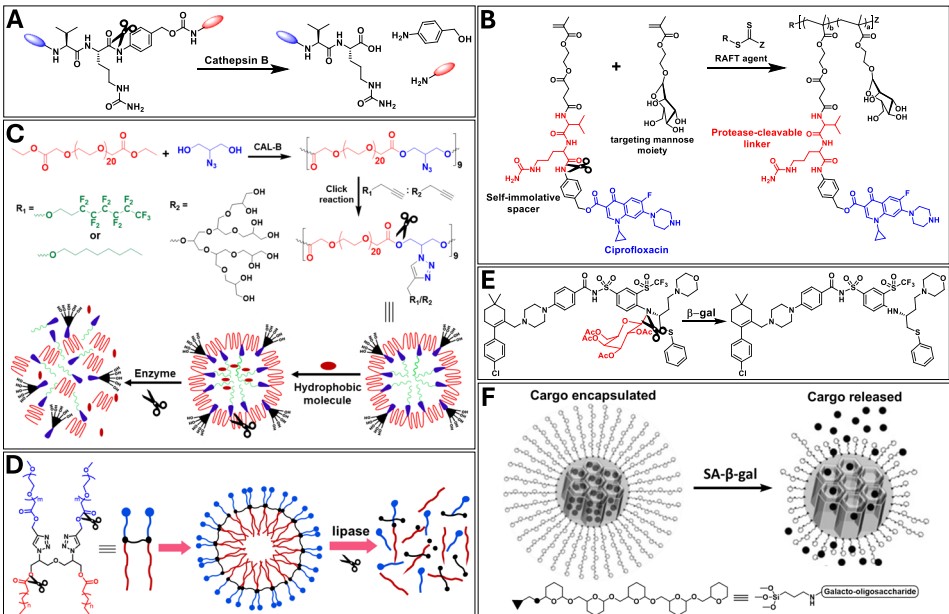

**Fig. 2 | Illustration of enzyme-responsive cleavable conjugations. A** Chemical structure of Val-Cit linker and its cleaving site. **B** Utilization of Val-Cit linkage in Cathepsin B cleavable polymeric prodrug formulation. **C** Synthesis, self-assembly and drug release behavior of CAL-B responsive amphiphilic copolymer. Reprinted with permission from ref. 56. Copyright 2016 MDPI. **D** Lipase-responsive non-ionic small amphiphiles' self-assembly and disassembly strategy. **E** Molecular structure of galactose-functionalized Navitoclax prodrug. **F** Mesoporous silica nanoparticles coated with galactose oligosaccharide to deliver payload in senescent cells. Reprinted with permission from ref. 68. Copyright 2012 John Wiley and Sons.

attention due to their ability to disassemble and release encapsulated cargo in the presence of these enzymes. Various strategies have been utilized for designing lipase responsive materials. One approach involves employing polymeric/dendritic systems with multiple lipase-cleavable linkages in their backbone, which upon cleavage, leading to material degradation[54]. Another approach entails incorporating lipase-cleavable linkages between hydrophilic and hydrophobic units of lipid-based amphiphilic structures[55]. In the presence of lipase e.g. Candida antarctica lipase B (CAL-B), cleavable linkages connecting hydrophobic and hydrophilic undergo cleavage, leading to the disassembly and subsequently release of the payload. Sharma et al. have developed lipase-responsive drug delivery nanocarriers using PEG and glycerol-based block copolymers for applications in cancer imaging and therapy (Fig. 2C)[54,56,57]. The polymerization occurs regioselectively via the primary hydroxyl groups of glycerol/azidoglycerol leaving the secondary hydroxyl/azide available for post-polymerization modifications and attaching drugs/bioactive molecules. Their study on encapsulated curcumin revealed stable polymeric nanostructures without enzyme while lipase incubation showed up to 90% release in 12 days[56]. They have also reported the synthesis of amphiphilic micelles with lipase-cleavable ester linkage for encapsulating and releasing dexamethasone, curcumin, nimodipine, and Nile red (Fig. 2D)[58,59]. Reported amphiphiles showed more than 80% release within 72 h while there was less than 10% release in control experiment without enzyme[55,60].

**Glycosidic linkages.** Glycosidase mediated cleavage strategy has shown promising advantages in design prodrugs and nanocarriers for the elimination of cancer and senescent cells from the biological system. Overexpression of senescence-associated β-galactosidase (SA-β-gal) activity is one of the most widely employed markers for senescence[61]. It has been demonstrated that β-gal could be exploited for the design of galactose-functionalized prodrug for the selective targeting of senescent cells[62]. The galactose-functionalized prodrug could be transformed into the parent active drug by the hydrolase activity of β-gal and subsequently destroy senescent cells, overcoming the limitations of current senolytic drugs. Several senolytic drugs such as gemcitabine, navitoclax, and

duocarmycin analog (JHB71A) have been modified into galactose-functionalized prodrugs namely SSK1, Nav-Gal (Fig. 2E) and GMD, respectively, showing increased selectivity towards senescent cells in aged mouse models[63–65].

Moreover, it has also been found that nanoparticles functionalized with galactose moiety preferentially release their payload in senescent cells[66,67]. Mesoporous silica nanoparticles coated with galactose oligosaccharide have been utilized to release payload in senescent cells driven by digestion of coated oligosaccharide by lysosomal β-gal (Fig. 2F)[68].

Numerous chromogenic and fluorescent probes, as well as imaging agents, have been designed to take advantage of the typically elevated β-gal levels in senescent cells[69,70]. Among them, 5-bromo-4-chloro-3-indolyl-β-D-galactopyranoside (X-Gal) is the most widely used colorimetric substrate for detecting β-gal activity in vitro. Upon enzymatic cleavage, X-Gal produces galactose and 5-bromo-4-chloro-3-hydroxyindole, which dimerizes to form a blue-colored indigo compound. Fluorescent probes such as $C_{12}FDG$, AHGa, SPiDER-β-gal and DDAO galactosides have been developed, where galactose cleavage results in fluorescence turn-on that was quenched initially[69,71]. Other relevant examples, such as SG1 and HMRef-βgal, have been reported to detect β-gal expressing cells in vivo, although not limited to cellular senescence[69,72].

The toolbox of cleavable linkers was further expanded by Bargh and coworkers where they utilized 3-$O$-sulfo-β-galactose linker to be sequentially cleaved by two lysosomal enzymes i.e., arylsulfatase A and β-gal[73]. The 3-$O$-sulfo-β-galactose linker was first hydrolyzed by arylsulfatase A to expose β-galactosyl ceramide, which was susceptible to β-gal, giving the ceramide metabolite. The dual enzymatic-cleavage linker strictly requires cleavage by both lysosomal enzymes for drug release. The motif incorporated α-HER2 antibody as the targeting ligand and was successfully bioconjugated to trastuzumab, resulting in a highly potent and selective antibody-drug conjugate.

**Glutathione (reductant)-responsive cleavable conjugations**
Glutathione (GSH), a common biological antioxidant that maintains cellular homeostasis in the most living organism, has a higher intracellular concentration of 2-10 mM as compared to the extracellular concentration of

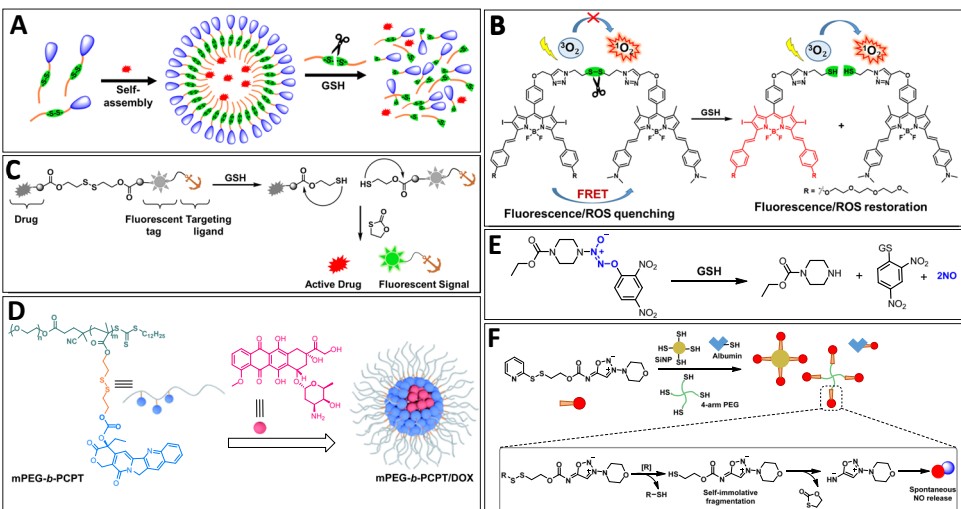

**Fig. 3 | Illustration of glutathione (reductant)-responsive cleavable conjugations.** **A** Schematic representation of GSH-responsive amphiphilic micelles. **B** Fluorescence restoration and ROS activation by GSH-mediated cleavage of -SS-linkage. **C** Schematic representation of -SS- containing fluorescently labeled targeting prodrug conjugates. **D** Amphiphilic prodrug copolymer containing multiple CPT and loaded with DOX for combination chemotherapy. Reprinted with permission from ref. 85. Copyright 2020 Royal Society of Chemistry. **E** Nitric oxide release from GSH-responsive nitric oxide prodrug. **F** Self-immolative multivalent nitric oxide prodrugs functionalized on SiNP, albumin and 4-arm-PEG. Reprinted with permission from ref. 104. Copyright 2022 John Wiley and Sons.

2–20 μM[74]. Moreover, the higher concentration of GSH is detected in cancer cells as compared to healthy cells, where this excess GSH promotes tumor progression[75]. The intracellular higher reducing environment of cancer cells is maintained by glutathione reductase and NADPH and depends upon the redox states of GSH/GSSG and NADPH/NADP. A reducing enzyme gamma-interferon-inducible lysosomal thiol reductase together with iron (ferrous) and cysteine reducing agent contributes to the higher reducing environment. In a reducing environment, glutathione concentration is higher than that of NADPH, where glutathione controls the cellular environment usually through the formation and degradation of disulfide bonds and by neutralizing excess ROS. Owing to this the concentration of glutathione is typically considered a major factor for the cellular reducing environment.

Thereby, incorporating chemical groups that respond towards concentration of GSH provides a feasible strategy for designing GSH-responsive materials. Disulfide and diselenide linkers are well-known examples to build nanoarchitectures that are stable enough during systemic circulation and in the extracellular environment but are readily cleaved by intracellular reducing agents[76,77]. GSH is known to cleave disulfide and diselenide bonds by binding to one cleaved thiol/selenol, forming another disulfide or sulfide-selenide linkage, liberating the other half of disulfide/diselenide as a thiol/selenol, thus being an efficient reducing agent.

**Disulfide linkage**. Braatz et al. investigated the GSH-triggered release of tyrosine kinase inhibitor anticancer drug Sunitinib encapsulated in biocompatible degradable polyglycerol-SS-polyester micelles as represented in Fig. 3A[78]. The disulfide linkage was selectively cleaved under the tumor microenvironment reducing condition, disturbing the hydrophilic-hydrophobic (amphiphilic) balance, and thus facilitating the sustained drug release inside the tumor cells. The anticancer efficacy of Sunitinib was found to enhance tenfold with micelles as compared to free drug.

Redox-sensitive liposomes were also employed for the treatment of orthotopic osteosarcoma. Feng et al. developed an osteosarcoma-targeting system grafted by bone and CD44 dual-targeting glutathione-responsive polymer[79]. The CD44 targeting moiety, hyaluronic acid, was first functionalized with bone targeting molecule, alendronate and then coupled to DSPE-PEG-COOH through GSH-cleavable disulfide (-SS-) linker to obtain amphiphilic lipid architecture. The synthesized architecture was post inserted into doxorubicin (DOX)-loaded preformed liposomes. Once the liposomes were internalized into tumor cells, intracellular GSH cleaved -SS-linkage and promoted the polymer shedding from the liposomes and caused the drug release. Attempts have also been made to create redox and pH-sensitive nanogels for the rapid release of DOX to triple-negative breast cancer cells[80]. The synthesized nanogels were based on dextran which was cross-linked with disulfide containing moiety (cystamine) for conferring sensitivity towards GSH. DOX was conjugated to oxidized dextran through an imine bond provided with pH sensitivity. The synthesized DOX-conjugated dextran-cystamine nanogels exhibited localized and sustained drug release triggered by rapid disassembly of nanogels in tumor cells via the cleavage of -SS- linkage. Mesoporous silica nanoparticles have been utilized for selective internalization in tumor cells with endosomal escape, where the particles were decorated by self-immolative GSH-cleavable linkers as smart gatekeepers to prevent premature release[81]. Further, the GSH cleavable linker was further attached to a recognition ligand to promote their specific accumulation in cancer cells.

The GSH-responsive behavior of disulfide linkage was also utilized in the construction of photosensitizer which is equipped with fluorescence resonance energy transfer (FRET) characteristic to be used in imaging-guided targeting photodynamic therapy[82]. Two carefully designed boron dipyrromethene derivatives were utilized as photosensitizing fluorescence and quencher probes, respectively, and linked to each other through a disulfide bond (Fig. 3B). Owing to the intramolecular FRET mechanism between photosensitizer and quencher, the activatable photosensitizer remains in a silent state. In the presence of GSH, the disulfide linkage was cleaved, inactivating the FRET, and activating the photosensitized to produce fluorescence and produced singlet oxygen efficiently. The GSH-responsive behavior of the photosensitizer was validated through in vitro and in vivo assays, where in vivo experiments with H22 tumor-bearing mice demonstrate strong tumor imaging.

Kim and co-workers successfully synthesized a variety of disulfide-linked tumor targeting fluorescent prodrug conjugates as represented in Fig. 3C[83]. A range of chemotherapeutic agents such as cisplatin, doxorubicin, paclitaxel, gemcitabine and camptothecin were attached to various fluorophores such as coumarin, cyanine, rhodol, BODIPY and naphthalimide, through a disulfide linkage to obtain glutathione-responsive drug-disulfide-fluorophore theranostic conjugates[83]. The tumor targeting ability of drug-disulfide-fluorophore conjugate could be enhanced by attaching a specific site-targeting ligands such as biotin, folate, galactose and RGD peptide which exhibit higher selectivity towards cancer cells over noncancer cells

due to their genetic signatures or owing to overexpression of a particular receptors on certain cancer cells. In one of the studies, anticancer drug camptothecin (CPT) was linked to naphthalimide fluorophore via disulfide-bond to afford a CPT-nanpthalimide prodrug conjugate[84]. The cleavage of the disulfide bond in prodrug conjugate by endogenous glutathione disrupted the internal charge transfer phenomena and induced a significant fluorescence shift together with release of attached drug. Furthermore, an RGD cyclic peptide was installed on the fluorophore head as a tumor-targeting agent. Conjugate was found to be more active in U87 cells than in C6 cells and released the drug within the endoplasmic reticulum of the cells while the control conjugate with non-cleavable linkage was found to accumulate mainly in mitochondria. A GSH-responsive amphiphilic prodrug copolymer (mPEG-*b*-PCPT) containing multiple CPT was synthesized and loaded with DOX for combination chemotherapy (Fig. 3D)[85]. Upon internalization into the tumor cells, GSH in the cytoplasm cleaves the disulfide bond within the polymer, triggering the CPT release and disassembly of the system which facilitated the immediate co-release of the DOX. The cellular uptake study displayed the effective internalization and GSH-responsive co-release of drugs into HepG-2 cells. MTT assay and cell apoptosis studies revealed significantly greater anticancer efficacy with mPEG-*b*-PCPT@DOX compared to the combination of free drugs, displaying excellent synergistic effect.

**Diselenide linkage.** Inspired by the redox-sensitive nature of disulfide linkage, diselenide linkage has been utilized for construction of GSH-responsive materials. Due to the large atomic radius and weaker electronegativity of selenium compared to sulfur, selenium containing compounds show a lower bond energy (C-Se bond 244 kJ/mol and Se-Se bond 172 kJ/mol) than sulfur containing compounds (C-S bond 272 kJ/mol and S-S bond 240 kJ/mol). Several studies have shown that the diselenide linkage containing materials such as hydrogels, polymers and prodrugs act as efficient redox-responsive drug delivery systems[86,87]. Hailemeskel et al. reported the synthesis of diselenide containing PEG-PCL-PEG triblock copolymer which formed uniform self-assembled nanoparticles in an aqueous medium[88]. In the presence of GSH, the PEG segments of the copolymer were cleaved from the micelles due to breakage of diselenide linkage, resulting into dissociation of micelles and subsequent drug release. Sun and coworkers studied and compared the impact of sulfur, selenium, and carbon linkages in paclitaxel-citronellol prodrug conjugates[89] Six different paclitaxel-citronellol prodrug conjugates containing either thioether, disulfide, selenoether, diselenide, carbon or carbon-carbon bond as a spacer have been synthesized and compared in terms of their self-assembly, redox-responsivity, drug loading and release behavior. Redox-responsive drug release from prodrug nano-assembly was investigated using dithiothreitol (DTT), a commonly used analog of GSH. Sulfur being much more active than selenium as an electron acceptor, undergoes reduction more easily and hence disulfide containing conjugates were more sensitive to reduction conditions than diselenide ones. The order of reductant induced drug release was found to be -S-S- > -Se-Se- > -S- > -Se-. The sulfur/selenium containing conjugates also exhibited $H_2O_2$ (a commonly used analog of ROS) responsive drug release. In the presence $H_2O_2$, the sulfur or selenium atoms in the linkers can undergo oxidation, forming sulfoxide/selenoxide or sulfone/selenone groups. This oxidation may enhance the hydrolysis of nearby ester bonds, leading to the release of the drug. Due to larger atomic radius and lower electronegativity of selenium compared to sulfur, selenium atoms are more sensitive to oxidation towards $H_2O_2$. The oxidant-responsive nature was found to follow the order: selenoether > thioether > diselenide > disulfide and displayed $H_2O_2$ triggered drug release.

**Other GSH-responsive linkages.** Chen and colleagues developed a near-infrared (NIR) fluorescent probe, CyA-cRGD, consisting of a cyanine dye as the fluorescence reporter linked to a tumor-targeting cRGD unit via a GSH-responsive nitroazo aryl ether group[90]. This probe exhibits

a highly selective response to GSH, enabling direct off-on fluorescence signaling for GSH detection.

Medical gas therapy which involves the treatment by the administration of therapeutic gases, such as nitric oxide, carbon monoxide and hydrogen sulfides, has emerged as a promising strategy for the treatment of cancer[91,92]. Several nitric oxide donor/prodrug molecules have been developed to treat cancer by regulating microenvironmental concentration of nitric oxide at target site. Reductant-activable nitric oxide prodrugs have been employed as smart nitric oxide surrogate for tumor microenvironment glutathione-responsive nitric oxide release[93–95]. [$O_2$-(2,4-Dinitrophenyl)1-[(4-ethoxycarbonyl)piperazin-1-yl]diazen-1-ium-1,2-diolate] popularly known as JS-K is an anticancer nitric oxide prodrug employed to generated nitric oxide under GSH rich tumor microenvironment (Fig. 3E). It reacts with GSH at physiological pH and generates 2 moles of nitric oxide. Double JS-K, a next generation analog of JS-K which generates 4 moles of nitric oxide per mole of compound, has also been shown to exhibit potent in vitro antiproliferative activity against human leukemia cells[96].

However, the delivery of JS-K and its analogs directly to the tumor site is quite challenging because of their limited stability and solubility properties. Nanoparticles incorporating nitric oxide prodrugs have been employed to achieve tumor targeted prodrug delivery and tumor microenvironment triggered nitric oxide release. Various drug carriers such as polymers[97] dendrimers[98] nanoparticles[99–101] and hydrogels[102,103] have been utilized as efficient drug delivery vehicles. Hu et al. developed nanoscale coordination polymer by coordinating nitic oxide prodrug with iron metal ions through simple precipitation process and ion exchange method[94]. Moreover, rapid nitric oxide release in presence of glutathione in tumor cells showed improved nitric oxide release thus avoiding side effects in other tissues. Kim et al. developed GSH-triggered self-immolative nitric oxide prodrug and functionalized it onto various materials such as albumin, 4-arm-PEG, and silica nanoparticles to improve lymphatic drainage and cancer cell uptake (Fig. 3F)[104]. The nanosized albumin-prodrug conjugate found to exhibit enhanced lymphatic drainage and displayed cytosol selective nitric oxide release.

## Oxidant-responsive cleavable conjugations

Reactive oxygen spices (ROS) are chemically reactive partially reduced metabolites of oxygen that play an important role in regulating myriads of physiological processes such as cell growth, proliferation, and signaling[105]. ROS are produced during cellular respiration where various oxidases take the electron liberated from membrane carriers such as coenzyme Q and cytochrome c and convert them into superoxide anions. Typical ROS consist of hydroxyl radical, superoxide anion, and hydrogen peroxide, and in addition one ROS type can convert into another via a series of reaction processes. The superoxide anions are converted into hydrogen peroxide by superoxide dismutase and xanthine oxidase in the cytosol and mitochondria. A balanced ROS level is essential for homeostasis, an increased level in certain pathological disorders can cause damage to cells by promoting the oxidation of cellular membranes, proteins, and nucleic acids[106]. Moreover, excessive intracellular ROS production is implicated in various pathological conditions including aging, cellular senescence, cancer, neurological and inflammatory diseases[5]. This has triggered interest in developing strategies for the creation of ROS-responsive functional materials. ROS-responsive materials can undergo physical and chemical degradation if they contain ROS-induced degradable moieties such as thioacetal and thioketals, aryl-boronic ester, oligoproline, or aminoacrylate.

**Thioacetal and thioketal linkages.** Thioketal-containing polymers are one of the most studied ROS-degradable materials[107]. Thioketal bond was found to be cleaved under several types of ROS including $H_2O_2$, hydroxyl radical, and superoxide to produce two thiol-containing fragments which then lead to material scission and degradation. Thioketal has been used as a functional cleavable linker to produce ROS-degradable materials for targeted drug delivery to cancer and senescent cells. Farokhzad and coworkers prepared polyprodrug (poly-mitoxantrone) having

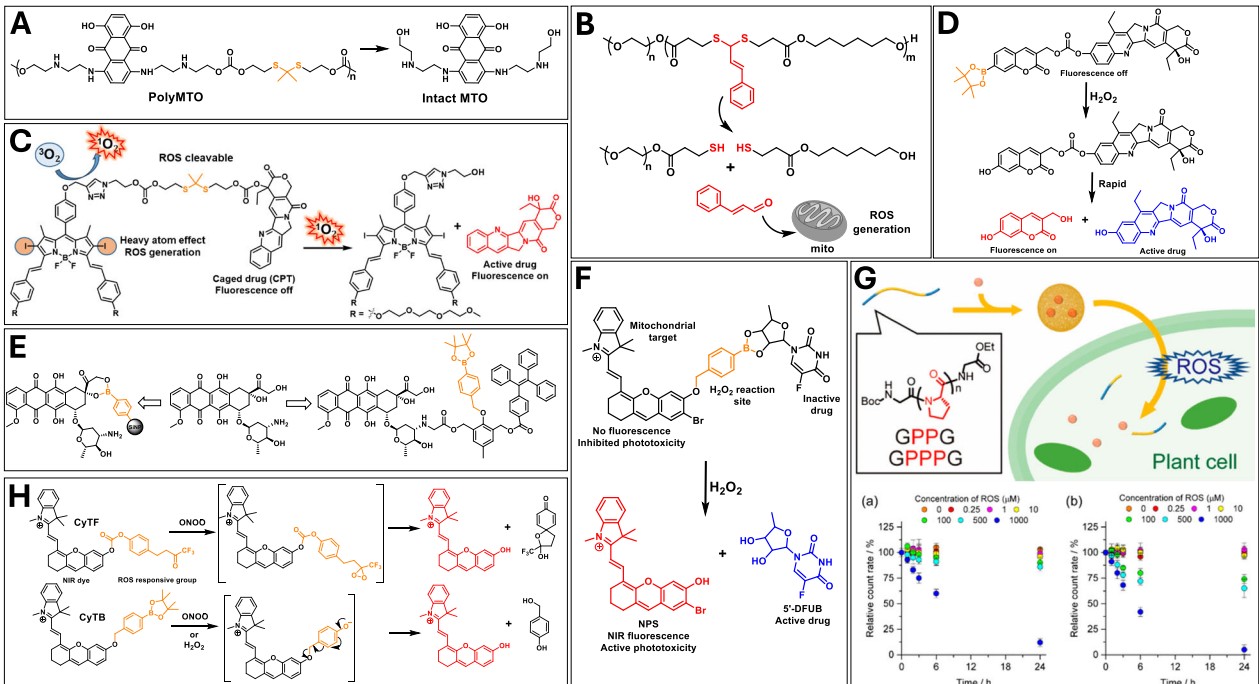

**Fig. 4 | Illustration of oxidant-responsive cleavable conjugations. A** Poly-mitoxantrone containing thioketal linkages and its cleavage under ROS conditions. **B** Cleavage of ROS-sensitive thioketal linkage and release of cinnamaldehyde from the polymer to trigger the mitochondria to regenerate additional ROS. **C** A red light-responsive prodrug BDP-TK-CPT containing ROS-labile thioketal linker. **D** H$_2$O$_2$-responsive theranostic prodrug containing coumarin and SN-38. **E** Chemical structures of prodrug construct of DOX. **F** A mitochondrial-targeting theranostic prodrug (PNPS) containing the drug 5'-deoxy-5-fluorouridine (5'-DFUR) attached to NIR photosensitizer (NPS) through a bisboronate group. **G** Peptide having more than two consecutive proline residues was observed to undergo cleavage in the presence of ROS generated by H$_2$O$_2$ in the presence of CuSO$_4$. Reprinted with permission from ref. [117]. Copyright 2020 American Chemical Society. **H** Chemical structures of CyTF and CyBA, and their RONS responsive cleavage.

ROS-cleavable thioketal linkers (Fig. 4A)[108]. Poly-mitoxantrone formed self-assembled nanoparticles in the presence of lipid-PEG and lipid-PEG-iRGD, which were employed for long blood circulation and tumor targeting features, respectively. In vivo efficacy of the nanoparticle after 18 days showed that there was less than 2-fold increase in tumor volume which was lower than the volume increase of mice treated with free mitoxantrone or nanoparticle without iRGD.

Xu et al. synthesized thioacetal-containing ROS-responsive polymeric micelles and additionally introduced cinnamaldehyde to trigger the generation of more ROS, further enhancing the ROS-responsive effect of system[109]. The intracellular ROS detection revealed that when the ROS-sensitive thioketal linkage cleaved in the presence of tumor microenvironment, the cinnamaldehyde moiety was released, triggering the mitochondria to regenerate additional ROS (Fig. 4B). Overall, the process synergistically enhanced the anticancer efficacy and achieved high drug delivery efficiency of encapsulated DOX.

A red light-activated prodrug, BDP-TK-CPT, was designed by linking the chemotherapeutic agent CPT to a boron dipyrromethene (BDP)-based photosensitizer through a ROS-sensitive thioketal linker (Fig. 4C)[110]. Since CPT is modified at its active site with a BDP-based macrocycle, the prodrug exhibits minimal toxicity in the absence of light. However, upon red light exposure, it efficiently produces ROS, inducing cell death through photodynamic therapy. Simultaneously, the ROS degrade the thioketal linker, releasing CPT and further enhancing localized cell death.

**Boronic ester linkage.** Among the various ROS-cleavable linkers, arylboronic ester is unique owing to its high selectivity towards H$_2$O$_2$. H$_2$O$_2$, peroxynitrite (ONOO$^-$) and hypochlorite anion (ClO$^-$) were found to react with boronic esters, oxidizing boron-carbon bond to hydroxy group. Recently, several arylboronic ester derivatives have been studied extensively to fabricate nanocarriers, prodrugs and imaging probes to release the parent compound in ROS-rich environment[111].

Metastatic tumors have been shown to exhibit elevated levels of ROS, including H$_2$O$_2$, supporting the idea that a prodrug activated by intracellular H$_2$O$_2$ could serve as an effective antimetastatic treatment. Kim and colleagues designed a theranostic prodrug incorporating an H$_2$O$_2$-responsive boronated ester, which triggers the activation of a fluorescent coumarin moiety and the release of the potent anticancer agent SN-38 (Fig. 4D)[112]. A similar strategy was employed to develop a theranostic prodrug construct of DOX and tetraphenylethene fluorophore (Fig. 4E)[113]. In H$_2$O$_2$ environment, reaction of boronic ester led to the separation of DOX and fluorophore, resulting in red emission of DOX and blue emission of fluorophore. Silica nanoparticles having phenyl boronic ester moiety at the surface were reported to load DOX covalently to overcome the low loading efficiency via physical adsorption (Fig. 4E)[114]. DOX was attached through its cis diol segment to generate boronic ester linkage, leading to nanoparticles covered with DOX. Nanoparticles were also functionalized with hyaluronic acid to increase their specificity towards CD44 receptors which are overexpressed in certain cancer cells and efficient release of DOX was observed in presence of H$_2$O$_2$ in CD44 containing Hep G2 cancer cells.

A mitochondrial-targeting H$_2$O$_2$-responsive theranostic prodrug (PNPS) was developed by conjugating 5'-deoxy-5-fluorouridine (5'-DFUR) to NIR photosensitizer (NPS) through a bisboronate group (Fig. 4F)[115]. In the presence of high concentration of H$_2$O$_2$ in tumor microenvironment, the bisboronate group is cleaved, resulting in activation of 5'-DFUR for chemotherapy and activation of NPS for NIR photodynamic therapy.

Zheng et al. reported ROS-responsive arylboronic ester-containing polymeric nanocarrier to deliver siRNA[116]. The siRNA was stabilized by hydrogen bonding, electrostatic interaction, and hydrophobic interactions and showed improved circulation stability and delivery efficiency. When nanocarriers encountered the overproduced ROS in the cancer microenvironment, the hydrophobic arylboronic ester converted into a hydrophilic carboxylic acid counterpart. This hydrophobic to hydrophilic shift reduced the hydrophobic stabilization and subsequently the formed

carboxylic acid interferes with hydrogen bonding and electrostatic interactions. This sequential complicated self-destruct process resulted in the efficient release of siRNA.

**Other oxidant cleavable linkages**. Peptide having more than two consecutive proline residues was observed to undergo cleavage in the presence of ROS, generated by $H_2O_2$ in the presence of $CuSO_4$ (Fig. 4G)[117]. Oligoproline-based amphiphilic peptides (named GPPG and GPPPG) were developed as nanocarriers for delivery of hydrophobic cargo to plant cells. The encapsulated cargo was observed to be released into the plant cell's cytosol in presence of ROS generated in chloroplasts by light.

Cheng et al. introduced NIR fluorescent probes, CyTF and CyBA, capable of distinguishing keloid fibroblast from normal thermal fibroblast (Fig. 4H)[118]. CyTF contains trifluoromethyl ketone group linked to a NIR hemicyanine dye, exhibiting selective oxidation mediated by $ONOO^-$. CyBA, on the other hand, utilized a boronic acid-based self-immolative group which can react with both $ONOO^-$ and $H_2O_2$. CyTF with higher specificity towards $ONOO^-$, demonstrating superior sensitivity in detecting stimulated fibroblasts, both in vitro and in a xenograft live mouse model.

## Hypoxia-triggered cleavable conjugations
Hypoxia is condition refers to the absence of enough oxygen level in cells and tissues that is required to sustain biological functions. Cancer cells can selectively be targeted by hypoxia-triggered mechanism because they are found to possess lower oxygen concentration of ~0.02–2% as compared to normal cells with oxygen concentration of ~2–9%. In hypoxia conditions, several enzymes including oxidoreductases and flavoproteins are specially activated. Several chemical conjugations such as azo linkage, indolequinone, trimethyl locked quinone and nitro-containing aromatic compounds are found to behave as hypoxia sensitive conjugations (Fig. 5). Hypoxia causes the upregulation of nitroreductase with 100–1000-fold higher concentration than normoxic cells.

**Azo linkage**. Verwilst et al. developed a hypoxia-cleavable prodrug **5A** having rhodamine attached through azo linkage[119]. Azo bond also acts as a fluorescence quencher owing to its electron withdrawing nature and rapid conformational change around the -N=N- bond upon photo-excitation. In hypoxic conditions, cleavage of azo bond allowed the activation of prodrug with the release of rhodamine thus switch-on signal enhancement. The prodrug exhibited notable in vitro inhibition against MDA-MB-231 and DU145 under 3% $O_2$ hypoxic conditions. Kiyose et al. developed a cyanine based NIR probe for imaging hypoxic cell environments and real-time monitoring of ischemia in live mice[120]. The probe's non-fluorescent state under normoxic conditions, attributed to a FRET mechanism between cyanine and BHQ3, transition to fluorescence in hypoxic environment due to quick reduction and loss of FRET. Piao et al. further improved this approach by directly conjugating the azo

group to rhodamine, which when cleaved under hypoxia, restore their fluorescence[121]. In vivo imaging studies demonstrated the capability of probe to detect hypoxia in a rat model of retinal artery occlusion. A hypoxia-responsive polymer conjugate **5B** was developed by coupling hyaluronic acid with BHQ3 moiety through azo bonds and encapsulated with DOX[122]. In vitro assessments of cytotoxicity and cellular uptake revealed the efficient release of DOX in hypoxia environments upon azo bond cleavage, leading to higher cytotoxicity than in normal conditions.

**Quinone moiety**. Quinone moiety has also been utilized to develop hypoxia-triggered prodrug molecules owing to its ability to undergo electronic rearrangement to cleave covalent bonds under hypoxic conditions[123]. CPT was attached to quinone propionic acid through an aminobenzyl alcohol linker, resulting in a hypoxia-sensitive prodrug **5C**[124]. The compound exhibited notable cytotoxicity in cells with elevated DT-diaphorase expression and enabled the simultaneous monitoring of the in vitro drug activation process. Indolequinone moiety has also been utilized to quench fluorescence of a conjugated system (probe **5D**)[125]. Under hypoxia environment, quinone moiety undergo rearrangement, releasing the fluorophore with an increase in fluorescence intensity.

**Nitro group**. Nitro-substituted aromatic compounds have also been used for the construction of hypoxia-activated bioimaging and therapeutic probes. Nitroaromatic compounds are efficiently reduced to corresponding amines by nitroreductase in the presence of NADH. Kumar et al. reported a hypoxia-sensitive prodrug conjugate **5E**, by the combination of anticancer drug SN38, cancer-targeting biotin, and hypoxia-sensitive self-immolative linker 4-nitrobenzyl, for the diagnosis and treatment of solid tumors[126]. Activation by nitroreductase under hypoxia conditions, caused the release of SN38 with high selectivity and cytotoxicity toward biotin-positive tumor cells. In vivo, studies further validated the accumulation of **5E** in solid tumor, and strong anticancer effect under hypoxia conditions. Nitro group is also known to quench fluorescence of the attached fluorophore because of its strong electron withdrawing nature, however fluorescence can be restored upon its reduction to amine. A series of nitro-containing hypoxia-responsive fluorescent probes have been developed. These probes, when activated by nitroreductase and NADH, demonstrated efficient and selective release of fluorescent molecules such as Nile Blue or resorufin[127,128]. A fluorescent probe containing Nile blue linked to p-nitrobenzyl moiety via carbamate linkage, showed spontaneous release of Nile blue under hypoxic environment[128].

## pH-responsive cleavable conjugations
pH cleavable materials are a class of materials that have breakable functional groups either in their backbone, end group, or side chain and are capable of exhibiting pH-dependent physiochemical behavior. Materials with pH-cleavable chemical linkages including as acetal, ketal, imine, hydrazine, and

**Fig. 5 | Molecular structures of various hypoxia-sensitive chemical probes.** These probes respond to hypoxia environment, enabling the release of active moieties for applications in tumor imaging and therapeutic targeting.

orthoester have been explored to utilize the pH gradient. The emergence of a variety of pH-cleavable materials to cure and diagnose different pathological conditions and diseases has been witnessed recently in nanomedicine. The lower tumor pH has been utilized as a potential endogenous stimulus for the fabrication of smart materials capable of imaging, diagnosing, and treating cancer cells selectively under an acidic tumor environment. Moreover, the acidic nature of intracellular components such as lysosomes and endosomes has also been utilized for the development of pH-sensitive nanocarriers designed for intracellular delivery of guests.

Different kinds of pH-cleavable nanocarriers utilizing pH difference between healthy and diseases cell have been reported for the target specific release of active pharmaceutical molecules. pH-cleavable nanocarriers constructed from polymers that transition between hydrophilic and hydrophobic states because of cleavage of acid-sensitive functional groups have widely been explored in the literature. Shifting from a hydrophilic to hydrophobic state results in decreased solubilization of the system while shifting from hydrophobic to hydrophilic causes improved solubilization, in both cases the nature of the nanocarriers in an aqueous environment changed and consequently leads cargo release.

**Schiff base linkage**. Polymers containing Schiff base linkages, where the hydrolysis of Schiff base cause drug release in acidic medium, have extensively been explored for the construction of pH-cleavable nanocarriers. Yu et al. utilized imine linkage to conjugate DOX to PLA-based polymer to construct a pH-degradable brush polymer. The pendent aldehyde groups of the PLA backbone were used to conjugate DOX to create a polymer-drug conjugate which formed self-assembled micellar aggregates in aqueous medium. The resulting self-assembled micelles demonstrated fast release kinetics under acidic medium[129].

Imine linkage has also been used to attach pH-responsive gatekeeper to create pore cap on the outer surface of the mesoporous silica nanoparticles with open-close pore transformation[130]. Since imine bonds hydrolyze under acidic conditions, the gatekeeper molecules such as PEG, polyglycerol, peptide, and carbohydrate could be severed from the surface at low pH thus opening the pores and releasing the loaded guest molecules[131–133]. Recently, Pan et al. developed polyglycerol-functionalized mesoporous silica nanoparticles for the co-delivery of DOX and tariquidar to eliminate cancer stem cells and overcome multidrug resistance in breast cancer stem cells[134].

The hydrazone functionality is another commonly employed Schiff base functionality to fabricate pH-cleavable drug delivery system. The hydrazone linkage is relatively stable and hydrolyzes very slowly at pH 7.4 while at low pH 5-6 in endosomal/lysosomal compartments the rate of hydrolysis is increased. Aryal et al. prepared self-assembled antitumor nanoparticles of sub-100 nm size comprising of polymer-cisplatin conjugates linked by hydrazone bonds[135]. Functionalization of hydrazide containing PEG-*b*-PLA polymer with Pt(IV) prodrug through ketone group afforded the polymer-drug conjugate where each cisplatin drug was linked to two polymer chain (Fig. 6A). The drug release rate from the nanoparticles was significantly faster at pH 5.0 and 6.0 than pH 7.4 as 50% release occurred in 4 and 6 h at pH 5.0 and 6.0, respectively as compared to 22 h at pH 7.4.

Hydrazone linkage has also been utilized in the construction of a turn on fluorescent probe that enabled direct monitoring of pH-mediated cleavage process[136]. The probe was based on polyglycerol scaffold which consisted of a fluorescent donor dye indocarbocyanine (ICC) and an acceptor indodicarbocyanine (IDCC) dye linked in proximity through a pH-labile hydrazone bond (Fig. 6B). The fluorescence signal of the donor dye is quenched by FRET between donor and acceptor until it is released by the cleavage of hydrazone linkage. The system exhibited monitoring of increasing fluorescence of indocarbocyanine dye upon release in real time.

Park et al. developed biotin-targeted drug delivery system with pH-mediated drug release profile[137]. An anticancer prodrug consisting of a targeting ligand (biotin) and DOX connected via hydrazine containing fluorescent quencher molecule (nitrobenzene) (Fig. 6C). Prodrug was found to be a promising candidate owing to its higher selectivity towards biotin receptor-positive HepG2 cells as compared to biotin receptor-negative WI-

38 cells. The DOX release was triggered by pH-mediated hydrazone bond cleavage, accompanied by an increase in fluorescence intensity that was initially quenched by the nitrobenzene unit. The results showed that the prodrug accumulated preferentially in biotin-positive HepG2 cancer cells and activated by low pH to release the drug with fluorescence enhancement. In a different study, hydrazone-based prodrug containing DOX and Gd$^{3+}$-texaphyrin moieties was developed as a theranostic drug delivery system[138]. The prodrug conjugate showed tumor-selective release of drug and allowed fate of system to be followed by two complementary imaging techniques, i.e., off-on fluorescence enhancement (fluorescent DOX unit) and a magnetic resonance imaging (paramagnetic Gd$^{3+}$-texaphyrin moiety). Further experiments showed that the conjugate was more extensively taken up by cancerous A549 and CT26 cells than the control non-cancerous NIH3T3 cells.

pH-responsive systems extend beyond drug delivery and play a significant role in theranostics, a field that combines therapeutic and diagnostic functions within a single platform, including chemotherapy and photo-thermal/photodynamic therapy. Polymeric micelles have emerged as a promising nanoplatform for cancer theranostics. Chen and colleagues developed pH-responsive polymeric micelles encapsulating DOX for dual-function aggregation-induced emission (AIE) imaging and chemotherapy[139]. They synthesized a novel zwitterionic copolymer, poly(MPC-co-FPEMA), via RAFT polymerization and modified it by conjugating tetraphenylethene (TPE), an AIE chromophore, through hydrazone bonds, forming PMPC-hyd-TPE. The resulting AIE-activatable copolymer self-assembled into spherical PC-hyd-TPE micelles, which were loaded with DOX. These zwitterionic micelles exhibited excellent physiological stability and minimal protein adsorption due to their stealthy phosphorylcholine (PC) shell. Under acidic conditions, the cleavage of hydrophobic TPE induced micelle swelling, as observed through size changes at pH 5.0. In vitro drug release studies demonstrated an accelerated release of DOX when the pH dropped from 7.4 to 5.0. Additionally, ex vivo imaging confirmed efficient drug accumulation and release in tumor tissues. Overall, these multifunctional micelles, featuring a nonfouling surface, AIE-based imaging, and pH-responsive drug release, show great promise as advanced theranostic systems for cancer treatment.

**Acetal and ketal bonds**. In addition to Schiff base, a wide variety of architectures utilizing acetal and ketal linkages as pH-cleavable functionalities have been utilized to construct pH-responsive drug delivery nanocarriers. pH-cleavable acetal and ketal linkages have been incorporated into the polymer backbone and used as a junction between hydrophobic and hydrophilic parts. They have also been used to attach drugs to polymer or nanoparticle backbone. Hu et al. synthesized a three-arm star-block polymer by the combination of mPEG and poly(ε-caprolactone) (PCL) linked through acetal groups (Fig. 6D)[140]. The synthesized star polymer was shown to form self-assembled micellar aggregates of spherical and cylindrical morphology. The DOX-encapsulated micelles demonstrated pH-mediated release kinetics and the in vitro experiment on HeLa cells showed systematic intracellular DOX delivery. Although, examples containing the Schiff base linkages in the side chain of the polymers are quite common but the polymers with backbone Schiff bond are very rare. A triblock copolymer PEG-PCL-PEG consisting of oxime linkages at the junctions of PEG and PCL was developed and found that the micelles formed by the polymeric system are promising pH-responsive drug carrier[141].

The acetal functionalities have also been used for grafting hydrophobic side chains to hydrophilic polymer backbone in an optimized manner in order to get an overall amphiphilic system which subsequently form self-assembled micelles in an aqueous medium. Upon low pH exposure, the acetal hydrolysis resulting in conversion of amphiphilic system to hydrophilic polymer again and subsequently significant breakage of micelles and release of the encapsulated guest. Particularly, trimethyoxybenzylidene acetal moieties attached to side chain of hydrophilic polymers such as polycarbonates[142,143] polypeptide[144] polyglycerol and PEG-*co*-glycerol, have

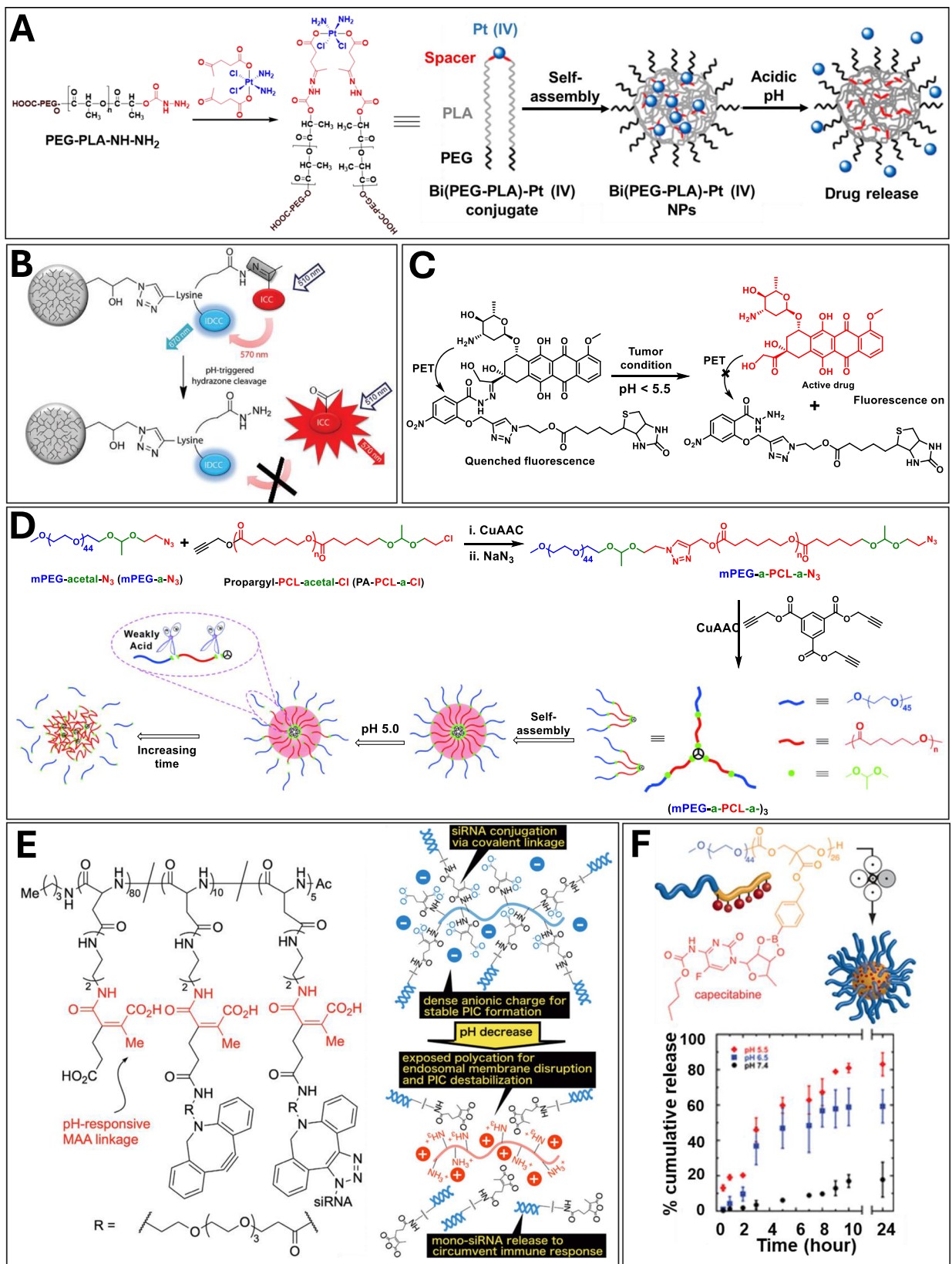

been utilized as efficient acetal-containing hydrophobic chains. Chen et al. developed pH-cleavable micelles having acid-cleavable trimethoxybenzylidene acetal moiety and investigated it for intracellular delivery of DOX[145]. The acetal linkages in the micelles remained stable at neutral pH but underwent rapid hydrolysis at pH 5.0 and 4.0, which resulted into

disassembly and drug release not only in mildly acidic endosomal compartments but also in the highly reducing cytoplasm and cell nuclei[146]. Haag and coworkers developed biodegradable polyglycerol nanogels where biodegradability was achieved by introducing the pH-cleavable acetal linkage at the junction between two different polyglycerol building blocks[147].

**Fig. 6 | Various pH-responsive materials. A** From synthesis to self-assembly and pH-mediated drug release behavior of polymer-cisplatin conjugate where each cisplatin drug was linked to two polymer chains via hydrazone bonds. Reprinted with permission from ref. 135. Copyright 2010 American Chemical Society. **B** Donor-acceptor based pH-cleavable fluorescent probe. Reprinted with permission from ref. 136. Copyright 2015 Royal Society of Chemistry. **C** A targeting prodrug consisting of biotin and DOX connected to fluorescent quencher nitrobenzene via hydrazone linkage. **D** Synthesis, self-assembly and dis-assembly behavior of acetal group containing three-arm star-block polymer. Reprinted with permission from ref. 140. Copyright 2015 Royal Society of Chemistry. **E** Chemical structure of polyaspartamide-siRNA complex and pH-triggered cleavage of cis-aconityl linkage to generate polycationic polymer and release mono-siRNA. Reprinted with permission from ref. 151. Copyright 2013 John Wiley and Sons. **F** A pH-responsive polymer-capecitabine conjugate based on boronic acid functionalized polycarbonate. Reprinted with permission from ref. 153. Copyright 2014 American Chemical Society.

The nanogels were found to degrade into low molecular weight fragments at low pH and released the encapsulated enzyme (asparaginase) with high specificity whereas no release was observed at pH 7.4.

**Orthoester linkages.** There are only a limited number of studies on poly(orthoester)-based pH-sensitive polymeric nanoparticles for targeted drug delivery because only a couple of synthetic strategies are available to prepare these types of polymers. The acid labile nature of orthoesters is more potent than esters, acetals and ketals and the synthetic methods are limited to the availability of monomers. Li et al. tried to address these challenges and synthesized a sugar-based poly(orthoester) polymer in which monomeric sugar units are connected by orthoester linkages in the backbone[148]. The synthesized system displayed a highly pH-responsive behavior with a half-life of 0.2, 0.4, and 0.9 h at pH 4, 5, and 6, respectively which displayed a very good stability at pH 7.4. The progress of hydrolysis at pH 6 was monitored by gel permeation chromatography which showed partial degradation of orthoester and resulted into sharp decrease in molecular weights and low molecular weight oligomers were observed with 2 h (Table 2).

**Cis-aconityl group.** *Cis*-aconityl group, an amide derivative of maleic acid, has also been employed as an acid-cleavable linkage for designing pH-responsive polymer-drug conjugates and drug carriers. The conjugation of amino group of DOX to cis-aconityl unit was found to be an effective strategy to develop pH-sensitive DOX prodrug as cis-aconityl linkage can easily be hydrolyzed in mildly acidic tumor microenvironment[149]. Zhu et al. reported polyamidoamine (PAMAM)-based DOX prodrug for effective tumor target[150]. In addition to DOX, PAMAM was functionalized with RGD peptide as a targeting ligand and PEG chain for prolonged blood circulation. pH dependent release studies displayed that the accumulative release of the DOX was increased with decrease in pH, where ~63, 16, 5 and 1.7% drug release was observed at pH 4.5, 5.5, 6.5, and 7.4, respectively. A similar acid-sensitive linker was utilized to trigger the charge reversal of cis-aconityl functionalized DOX prodrug at acidic pH and subsequently release DOX that was attached to the positively charged PEG-acrylate copolymer though synergetic electrostatic and hydrophobic interactions[149]. Approximate 80% drug release was observed within 12 h at pH 5.0 due to the synergetic effect of hydrolysis of cis-aconityl linkage leading to charge reversal of cis-aconityl functionalized DOX and further protonation of PEG-acrylate copolymer. Cis-aconityl linkages were utilized for the construction of poly-aspartamide based siRNA delivery system for programmed siRNA release in the cell interior (Fig. 6E)[151]. The designed system destabilized in endosome upon cleavage of cis-aconityl at low pH, leading to endosomal disruption with regenerated parent polycation and releasing mono-siRNA to circumvent immune system. The designed polyaspartaminde-siRNA conjugate displayed significant growth inhibition of cancerous cells than mono-siRNA/polyaspartamide (a non-covalent control). A similar strategy of charge reversal was utilized to trigger the charge reversal of cis-aconityl functionalized nanogel system at low pH and subsequently release DOX that was bind to nanogel surface through electrostatic interactions[152]. These cis-aconityl systems displayed same mechanism of action, where the amide bond is cleaved by carboxylic group in acidic environment.

Battogtokh et al. developed another pH-responsive theranostic agent incorporating a porphyrin-based photosensitizer, pheophorbide-a (PheoA), conjugated to bovine serum albumin (BSA) via a cis-aconityl linker[153]. To enhance targeting efficiency, the conjugate was further modified with polyethylene glycosylated folate. Self-assembly of BSA-c-PheoA and folate (FA)-BSA-c-PheoA in a 2:1 ratio resulted in nanoparticles with an average hydrodynamic diameter of 121.47 ± 11.60 nm. Release studies demonstrated that the photosensitizer was released 4.5 times faster at pH 5.0 than at pH 7.4 after 24 h of incubation. Cellular uptake experiments confirmed that FA-BSA-c-PheoA nanoparticles were efficiently internalized by B16F10 and MCF7 cancer cells. In vitro phototoxicity assays showed that FA-BSA-c-PheoA nanoparticles exhibited greater anticancer activity compared to BSA-c-PheoA nanoparticles alone. Additionally, in vivo bioimaging revealed that FA-BSA-c-PheoA nanoparticles accumulated significantly in tumor tissues compared to free PheoA. These findings suggest that FA-BSA-c-PheoA nanoparticles hold strong potential as theranostic agents for both photodynamic therapy and photo diagnosis of cancer.

**Other pH-cleavable linkages.** A polymer-drug conjugate based on boronic acid-functionalized polycarbonate for the pH-responsive delivery of diol-containing drug, capecitabine, was reported (Fig. 6F)[154]. Polymer-drug conjugate possessing amphiphilic behavior, was able to self-assemble into supramolecular nanostructures. The pH-mediated cleavage of boronic ester and subsequent release of active cargo can be broadly applied to various diol- and catechol-containing molecules, highlighting the versatility of these polymers for drug delivery applications.

## Challenges
This review critically analyses important biochemically triggered chemical conjugations that hold exciting potential to revolutionize modern medicine. However, their clinical applications face several significant challenges. Designing chemical conjugations that exclusively target specific pathological areas remains a complex endeavor. Introducing cleavable linkages into optimized systems often alters activity profiles, while chemical building blocks used in conjugations may prove incompatible with the intricacies of biological systems, leading to discouraging outcomes in clinical trials. Additionally, maintaining the activity of biomolecules such as antibodies, cargo, proteins and fluorescent tags is particularly challenging. Determining the optimal linker length is another obstacle, as it directly impacts the proximity and likelihood of ligand-target interactions. Lastly, both the designed systems and their by-products may pose toxic effects on living cells, raising critical safety concerns.

## Conclusion
Biochemically triggered chemical conjugations have emerged as a versatile platform in drug delivery, prodrug activation, imaging, and theranostics. They enable the design of sophisticated drug carriers for site-specific release, minimizing side effects and improving treatment outcomes. Prodrug activation using these conjugations enhances drug stability and targeted delivery, reducing off-target effects. Imaging benefits from highly specific contrast agents for accurate visualization of pathological sites, leading to improved patient care. Theranostics offers personalized medicine, combining diagnosis and treatment. Challenges remain, requiring further research for successful clinical translation.

**Table 2 | Important class of cleavable systems utilizing cleavable conjugations for targeted cargo delivery**

| | Cleavable systems | Characteristics | Cargo loading and releasing strategy |
|---|---|---|---|
| **Polymer/ Hydrogel** | 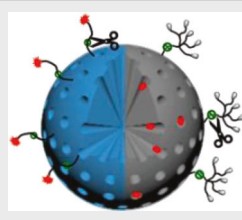 | • Different material such as dendrimer, polymersome, nanoemulsion, nanoaggregate and hydrogel can be synthesized by different methodologies.<br>• Useful for delivery of hydrophobic and hydrophilic payloads.<br>• Size, composition, stability, responsiveness and surface charge, together with loading efficacy and release kinetics can be precisely controlled.<br>• Highly biocompatible and water soluble/dispersible with known in vivo clearance mechanism | • Payload is usually linked to backbone via stimuli-responsive covalent linkage which can be cleaved in diseased microenvironment, giving active agent. |
| **Rigid Nanoparticles** | 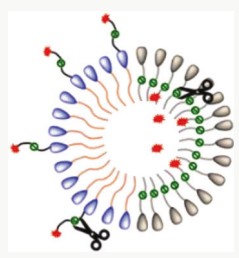 | • Can be precisely formulated into various sizes, shapes, and geometries.<br>• Such systems have qualified for applications such as diagnostics, imaging and photothermal therapies.<br>• They have high drug loading efficiency.<br>• Nanostructures are highly stable and not concentration dependent.<br>• Have a very low polydispersity index.<br>• They can hold small drug molecules to large polymers and even relatively large antibodies. | • Their hollow counterpart can be detached/cleaved in the physiochemical conditions thereby releasing the cargo.<br>• Cargo can be attached to the surface or loaded in the interior or surface. Physically loaded cargo is protected from the outer environment through attached surface coating or gatekeeper molecules. |
| **Lipid Nanoparticles** | 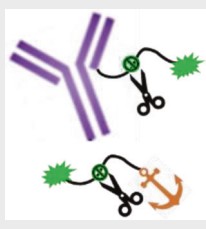 | • Lipid-based NPs can be easily modulated to control their physicochemical characteristics. That's why these are the most common classes of FDA-approved nanomedicines.<br>• Easily transformed into different nano-assemblies such as micelles, liposomes, cylindrical micelles, and sheets by the interplay of hydrophilic and hydrophobic units.<br>• They are truly biomimetic and are highly compatible with biological systems.<br>• Suitable for hydrophobic payloads. | • Most of such systems self-assemble under specific conditions while wrapping the cargo inside. Under the physiochemical changes of the diseased organ, intermediate linker cleaves to release the cargo. |
| **Pro-Drug** | | • Mainly contains three components: cytotoxic agent, targeting moiety and chemical linker, and in some cases fluorescent probe. Cleavable linkers play a key role in the success of antibody-drug conjugates and targeting prodrugs.<br>• Highly specific to diseased tissue.<br>• Usually stable in blood circulation. | • Cargo is attached through covalent linkage. In these systems, inherent properties of physiological conditions are utilized to selectively release cytotoxins. |

Cargo is either attached covalently to carrier scaffold through cleavable linker or loaded non-covalently.

## Future Perspective

As we advance towards personalized and targeted healthcare, the future of biochemically cleavable conjugations offers immense promise for transforming biomedicine. In targeted drug delivery, conjugation strategies designed to respond to specific molecular disease signatures could enable highly effective and personalized treatments for conditions such as cancer, autoimmune diseases, and chronic infections. In the realm of personalized medicine, a deeper understanding of disease mechanisms is poised to integrate biochemically cleavable conjugations into individualized treatment plans, enhancing therapeutic outcomes while minimizing adverse reactions.

Advanced imaging and diagnostics stand to benefit from conjugates with biochemically cleavable linkers that remain inactive until encountering specific biological triggers, improving specificity, reducing background noise, and enabling non-invasive, highly sensitive disease detection. The emergence of multimodal imaging agents capable of detecting multiple disease markers may further facilitate earlier and more accurate diagnoses. Theranostics, which merges therapeutic and diagnostic functionalities into a single entity, is another area of promise, as biochemically cleavable conjugations can enable real-time monitoring of treatment efficacy while delivering therapeutic agents to target sites. This approach allows clinicians to make data-driven decisions, optimizing treatment procedures based on patient responses. Finally, innovative conjugation strategies in combination therapies could enable precise and sequential drug release, improving treatment outcomes and reducing drug resistance.

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

## Acknowledgements

This project received support from the Medical Research Council (MRC), UK, through an MRC Career Development Award to HP (Grant no: MR/T030968/1) and a UCL TRO Award (Grant no: MRC IAA 2021 UCL MR/X502984/1) at University College London, UK. H.P. also would like to thank for EU H2020 Marie Sklodowska-Curie Individual Fellowship (Grant no: 706694).

## Author contributions

B.P.: Conceptualization, writing original draft, supervision, designing figures, reviewing and editing the draft, and literature search. S.A.: Conceptualization, writing original draft, preparing figures, literature search. B.R.: Writing original draft, preparing figures, literature search. Y.P.: Writing sections. J.T.: Critical review. Z.H.: Critical review. H.K.P.: Conceptualization, supervision, writing original draft, reviewing and editing the draft.

## Competing interests

The authors declare no competing interests.
