## [Transparent Peer Review file · Communications Chemistry]

Towards precision medicine using biochemically triggered cleavable conjugation

Corresponding Author: Professor Hirak Patra

Version 0:

Reviewer comments:

Reviewer #1

(Remarks to the Author)

This review explores the critical role of biochemically cleavable conjugations within intrinsically stimuli-responsive architectures. Pathological conditions and tissue microenvironment changes are defined by unique biochemical characteristics such as shifts in redox potential, pH variations, hypoxia, reactive oxygen species (ROS), and the overexpression of specific catalytic proteins or enzymes. Understanding these features is vital for designing advanced cleavable bio-engineered systems tailored for biomedical applications. This work offers valuable insights for advancing personalized medicine, although it contains some minor inaccuracies.

1. In Figure 3B on page 17, the boron dipyrromethene derivatives used as photosensitizers appear unusual. Upon reviewing the original publication, it seems an extra iodine substitution was mistakenly included and should be removed.
2. In Figure 4C on page 21, the image appears to be directly cut from the original publication and looks rough and unclear, with some text on the left edge being obscured.
3. Similarly, some figures are overly blurred, such as Figure 4H, which remains difficult to discern even when zoomed in. I recommend improving the resolution to make the figure clearer.
4. In Figure 4F on page 21, there is some confusion regarding the labeling. The PNPS label currently shown on the right should instead be placed on the yellow portion on the left. Additionally, the blue portion represents the anticancer drug (5'-DFUR) and should be labeled as an inactive drug when not released.
5. Some suggested recent references on disulfide bond: *Macromolecules* 2024, 57 (6), 2858-2867, *Angewandte Chemie* 2021, 133 (38), 21201-21207, *J. Am. Chem. Soc.* 2021, 143, 18446–18453.

Reviewer #2

(Remarks to the Author)

In the review article titled as 'Clever Chemistry for Translation: Biochemically Triggered Cleavable Conjugation²', authors have compiled the basic phenomena for local-regional microenvironment in biological system and the background about stimuli-responsive cleavages. After that, fundamental chemical principles for designing smart therapeutics that is triggered with the variation in physiological parameters. In this direction, almost all of cleavable chemical functionalities and their applications in nano-medicine approaches were properly enlightened. In my opinion, this timely and comprehensive review articles should be considered as a potential article in the journal after considering following points:

- Resolutions of some figures, especially combined figs, are not acceptable not only for printing quality but also for readability
- In the section 4, a classification based on basically chemical functional groups was compiled. Herein, a new title to cover polymeric degradation mechanisms such as erosive, bulky, and scissive etc should be added although those mechanisms follow the chemical reactions, which already summarized in respective schemes. By this title, it would be helpful to figure out how the degradation mechanism plays a major role in therapeutic activity beside the chemical cleavage reactions.
- Section 5- Challenges should be rewritten as a comparative literal way instead of bulleted sentences.
- Similarly, Section 7-Future Perspective should be summarized after emphasizing promising key areas as bulleted paragraphs while figuring out the their potential impacts in those fields.

Re: Subject: Revision of manuscript (COMMSCHEM-24-0611)

Reviewer #1 (Remarks to the Author):

This review explores the critical role of biochemically cleavable conjugations within intrinsically stimuli-responsive architectures. Pathological conditions and tissue microenvironment changes are defined by unique biochemical characteristics such as shifts in redox potential, pH variations, hypoxia, reactive oxygen species (ROS), and the overexpression of specific catalytic proteins or enzymes. Understanding these features is vital for designing advanced cleavable bio-engineered systems tailored for biomedical applications. This work offers valuable insights for advancing personalized medicine, although it contains some minor inaccuracies.

1. In Figure 3B on page 17, the boron dipyrromethene derivatives used as photosensitizers appear unusual. Upon reviewing the original publication, it seems an extra iodine substitution was mistakenly included and should be removed.

Response: Thank you for noticing this discrepancy. We have corrected the structure in the revised manuscript.

2. In Figure 4C on page 21, the image appears to be directly cut from the original publication and looks rough and unclear, with some text on the left edge being obscured.

Response: We are grateful that the reviewer has pointed out the figure quality. We have redrawn the structure in ChemDraw to improve the quality. We hope they now meet the standards required for Communications Chemistry.

3. Similarly, some figures are overly blurred, such as Figure 4H, which remains difficult to discern even when zoomed in. I recommend improving the resolution to make the figure clearer.

Response: Thank you for pointing this out. We have redrawn the structures (Figure 4A-4F & 4H) in ChemDraw to improve the quality.

4. In Figure 4F on page 21, there is some confusion regarding the labeling. The PNPS label currently shown on the right should instead be placed on the yellow portion on the left. Additionally, the blue portion represents the anticancer drug (5'-DFUR) and should be labeled as an inactive drug when not released.

Response: We have incorporated the suggested changes in the revised manuscript.

5. Some suggested recent references on disulfide bond: Macromolecules 2024, 57 (6), 2858-2867, Angewandte Chemie 2021, 133 (38), 21201-21207, J. Am. Chem. Soc. 2021, 143, 18446-18453.

Response: The suggested references have been incorporated.

Reviewer #2 (Remarks to the Author):

In the review article titled as 'Clever Chemistry for Translation: Biochemically Triggered Cleavable Conjugation², authors have compiled the basic phenomena for local-regional microenvironment in biological system and the background about stimuli-responsive cleavages. After that, fundamental chemical principles for designing smart therapeutics that is triggered with the variation in physiological parameters. In this direction, almost all of cleavable chemical functionalities and their applications in nano-medicine approaches were properly enlightened. In my opinion, this timely and comprehensive review articles should be considered as a potential article in the journal after considering following points:

- Resolutions of some figures, especially combined figs, are not acceptable not only for printing quality but also for readability

Response: We are grateful that the reviewer has pointed out the figure quality. We have redrawn the structure in ChemDraw to improve the quality. We hope they now meet the standards required for Communications Chemistry.

- In the section 4, a classification based on basically chemical functional groups was compiled. Herein, a new title to cover polymeric degradation mechanisms such as erosive, bulky, and scissive etc should be added although those mechanisms follow the chemical reactions, which already summarized in respective schemes. By this title, it would be helpful to figure out how the degradation mechanism plays a major role in therapeutic activity beside the chemical cleavage reactions.

Response: Thank you for your thoughtful feedback and suggestions regarding Section 4 of our manuscript. We greatly appreciate your efforts to enhance the quality of our work.

Regarding your suggestion to include a separate subsection on polymeric degradation mechanisms such as erosive, bulky, and scissive processes, we agree that these mechanisms are critical to therapeutic applications. However, we believe they are not isolated processes but operate in synergy with the chemical cleavage reactions already discussed. For example, the erosive release of a polymer coating can expose chemically cleavable linkages, initiating a cascade of mechanisms that optimize therapeutic delivery.

We have emphasized these synergistic interactions throughout the manuscript and included relevant citations (133-135, 138, 148 & 150) to substantiate our discussion. Adding a separate subsection could fragment the narrative and detract from the integrated view we aimed to present. We hope this clarifies our approach. If further elaboration is required, we are happy to revise the manuscript accordingly. Thank you again for your valuable input and support.

- Section 5- Challenges should be rewritten as a comparative literal way instead of bulleted sentences.

Response: Thank you for the valuable suggestion. We have carefully rewritten section 5 (Challenges) in the revised manuscript as follow:

This review critically analyzes various biochemically triggered chemical conjugations that hold exciting potential to revolutionize modern medicine. However, their clinical applications face several significant challenges. Designing conjugations that exclusively target specific pathological areas remains a complex endeavor. Introducing cleavable linkages into optimized systems often alters activity profiles, while chemical building blocks used in conjugations may prove incompatible with the intricacies of biological systems, leading to discouraging outcomes in clinical trials. Additionally, maintaining the activity of biomolecules such as antibodies, cargo, proteins, and fluorescent tags is particularly challenging. Determining the optimal linker length is another obstacle, as it directly impacts the proximity and likelihood of ligand-target interactions. Lastly, both the designed systems and their by-products may pose toxic effects on living cells, raising critical safety concerns.

- Similarly, Section 7-Future Perspective should be summarized after emphasizing promising key areas as bulleted paragraphs while figuring out their potential impacts in those fields.

Response: Thank you for the valuable suggestion. We have carefully rewritten section 5 (Challenges) in the revised manuscript as follow:

As we advance toward personalized and targeted healthcare, the future of biochemically cleavable conjugations offers immense promise for transforming biomedicine. In targeted drug delivery, conjugation strategies designed to respond to specific molecular disease signatures could enable highly effective and personalized treatments for conditions such as cancer, autoimmune diseases, and chronic infections. In the realm of personalized medicine, a deeper understanding of disease mechanisms is poised to integrate biochemically cleavable conjugations into individualized treatment plans, enhancing therapeutic outcomes while minimizing adverse reactions.

Advanced imaging and diagnostics stand to benefit from conjugates with biochemically cleavable linkers that remain inactive until encountering specific biological triggers, improving specificity, reducing background noise, and enabling non-invasive, highly sensitive disease detection. The emergence of multimodal imaging agents capable of detecting multiple disease markers may further facilitate earlier and more accurate diagnoses. Theranostics, which merges therapeutic and diagnostic functionalities into a single entity, is another area of promise, as biochemically cleavable conjugations can enable real-time monitoring of treatment efficacy while delivering therapeutic agents to target sites. This approach allows clinicians to make data-driven decisions,

optimizing treatment regimens based on patient responses. Finally, innovative conjugation strategies in combination therapies could enable precise and sequential drug release, improving treatment outcomes and reducing drug resistance.

With the revision, I strongly believe that it will be interest to a wider scientific community who are working on smart cleavable systems includes chemists, biomedical researchers, and students in the fields of chemistry, biochemistry, and materials science. Specifically, clinicians and pathologist would be interested in underpinning the mechanism of precision medicine and targeted drug delivery alongside developing new diagnostic approaches. We believe the regulatory bodies and policymakers will have a comprehensive overview and critical insights on evaluating safety and ethical considerations for future clinical use. Pharmaceutical and biotechnology industries would be interested to consider as a comprehensive toolbox in developing feasible and translational drug delivery systems and diagnostic approaches. Therefore, this review will attract interest across academia, industry, and healthcare sectors, with potential applications in future drug development with improved therapeutic outcomes.

With very best regards,

Hirak K Patra, PhD, FHEA, FIBMS, FRSB
Professor of Precision Nanosystems and Advanced Therapeutics
MRC Fellow (CDA), Medical Research Council UK